# Information sharing and channel structure in e-commerce supply chain considering data-driven marketing

**Feifei Han**[1], **Mei Wang**[1]*, **Zhengze Wu**[2]

1 School of Management, Wenzhou Business College, Wenzhou, PR China, 2 School of Information Management and Engineering, Shanghai University of Finance and Economics, Shanghai, PR China

* hualingse0211@163.com

## Abstract

E-tailers such as Amazon and Tmall can accurately recognize consumer interest in product categories and grasp current consumer trends through product recommendation algorithms and data-driven analysis. In this study, we develop a game-theoretic model to investigate the encroachment and information sharing decisions considering data-driven marketing (DDM). Our outcomes reveal that the manufacturer has the incentive to reduce the wholesale price to incentivize the e-tailer to increase the DDM effort when the spillover effect is high and the marginal cost is low. When the manufacturer encroaches on the direct channel, the e-tailer may share information if the spillover effect is low. Moreover, we derive the conditions under which the manufacturer encroaches on the direct channel. Lastly, we extend the model to the case where the agency channel exists, and we find that the manufacturer should select the agency channel only if the marginal cost of DDM effort is moderate and the commission rate is low. Our study provides useful insights for managers to understand and make channel choice and information sharing decisions in the e-commerce supply chain.

## 1. Introduction

### 1.1. Background and Motivation

With the rapid development of big data and mobile technology, consumers can easily browse product information and order through the e-tailer (https://www.collinsdictionary.com/us/dictionary/english/e-tailer). According to Statista, the global e-commerce market reached approximately $6 trillion in 2024, and is expected to reach $8 trillion by 2028 (https://www.statista.com/statistics/379046/worldwide-retail-e-commerce-sales/). In China, the total retail sales of social consumer goods were 47.1 trillion yuan in 2023, and online retail sales accounted for 27.6% of total retail sales (https://www.stats.gov.cn/english/PressRelease/202402/t20240201_1947119.html). In the

**Data availability statement:** All relevant data are within the paper and its Supporting Information files.

**Funding:** This research was supported by Zhejiang Provincial Regular Subjects of Philosophy and Social Science Planning (25NDJC104YB).

**Competing interests:** The authors declare that they have no known competing financial interests or personal relationships that could have appeared to influence the work reported in this paper.

United States, e-commerce sales reached $1.2 trillion in 2024, accounting for 22.7% of total retail sales, and are expected to grow to $1.6 trillion by 2028 (https://www.retaildive.com/news/online-retail-sales-increase-over-1-trillion-report/721564/). In 2024, over 2.7 billion people were expected to buy goods and services online (https://www.oberlo.com/statistics/how-many-people-shop-online). The high demand in the online market has attracted many manufacturers to distribute their products through online retail platforms such as Amazon, JD, and Tmall. For manufacturers who distribute the products through online retail platforms, the reselling channel is the only choice (e.g., in the early stages of Amazon and JD). The supplier is distant from the end consumers and lacks the flexibility to set prices in the reselling channel. In practice, a phenomenon known as supplier encroachment emerges, whereby some manufacturers choose to retail products in the end market through the direct online sales channel in addition to wholesaling products to the e-tailer, thus expanding the demand in the market. For example, Chinese smartphone maker OnePlus whole-sales the phones to JD and sells directly to end-consumers at the online OnePlus store (https://www.oneplus.com/cn). Also, Lee Kum Kee, a Hong Kong-based food company, wholesales the products to HKSuning.com and sells directly to end-consumers at its online store (https://shop.lkk.com/).

The manufacturer is distant from the end-consumer market and lacks information about market demand, whereas the e-tailer has more information about market demand because of the massive transactions and the closer contact with consumers. The e-tailer has access to a vast amount of consumer transaction and behavior data and can forecast downstream market demand. In addition, information technology can help e-tailers share data more effectively, and empirical studies show that information sharing improves the supply chain's ability to respond to uncertain market demand [1]. Meanwhile, e-tailers can accurately grasp current consumer trends through data-driven analysis. E-tailers can describe, predict, and analyze consumer behavior to locate the consumption points that most stimulate consumers' willingness to buy and realize personalized recommendations and precision marketing, thus improving consumer utility [2]. For example, Tmall has created a series of targeted marketing programs for Shiseido based on data and content and established a Shiseido-specific database (https://www.nbd.com.cn/rss/toutiao/articles/1338127.html). Amazon's "Selling Coach" allows merchants to grow their sales profitably on Amazon.com by tracking key metrics such as sales, traffic, and conversions (https://www.sellerapp.com/amazon-selling-coach.html). The purpose of all DDM efforts is to enable the e-tailer to expand the market demand and to increase the profit, and the e-tailer generally faces channel selection and information sharing decisions. The system-wide combined effect of the spillover effect and the marginal cost of DDM effort on the information sharing and encroachment decisions remains unclear.

## 1.2. Research questions and contributions

There is no existing theory to shed light on the interplay of information sharing and encroachment decisions in the e-commerce supply chain considering DDM effort.

We hope to bridge this gap by building analytical models to study the matter. We analyze four scenarios considering the decisions of the manufacturer and the e-tailer: no direct channel and no information sharing (NN), no direct channel and information sharing (NS), direct channel and no Information sharing (EN), and direct channel and information sharing (ES). We ask the following research questions.

1. What are the manufacturer's and e-tailer's equilibrium decisions (such as wholesale price, selling quantity, and DDM effort) under different scenarios?

2. What is the incentive for the e-tailer to share demand information with the manufacturer?

3. How do information sharing and the cost of DDM effort impact the manufacturer's encroachment decision of whether or not to add the direct channel to an existing reselling channel?

To study these questions, we develop a game-theoretic model in which one manufacturer has an existing reselling relationship with an e-tailer. When the manufacturer successfully encroaches, the manufacturer adopts both the reselling and direct channels to distribute products. To be consistent with the practices of e-tailers like Amazon and JD, we assume that the e-tailer exerts DDM effort to increase the market demand, and the e-tailer's DDM effort can also positively affect the demand in the manufacturer's direct channel. For scenario NN, the manufacturer decides on the wholesale price based on the prior beliefs, and then the e-tailer decides on the selling quantity and DDM effort based on the market demand information. For scenario EN, the manufacturer decides on the wholesale price in the reselling channel and the selling quantity in the direct channel based on prior beliefs. For scenario NS and scenario ES, the manufacturer and the e-tailer make decisions based on the market demand information. Our models generate several novel insights. It rigorously analyzes the interplay of information sharing and channel selection decisions and reveals how the spillover effect and the cost associated with DDM effort jointly determine channel structure and operational decisions. Moreover, it highlights the critical role of channel structure and information availability in shaping the e-tailer's incentive to invest in DDM activities.

## 1.3. Major findings and paper structure

Our analysis reveals that the manufacturer has the incentive to reduce the wholesale price to incentivize the e-tailer to increase the DDM effort when the spillover effect is large and the marginal cost of DDM effort is low. When the manufacturer encroaches on the direct channel, the e-tailer may share information if the spillover effect is low. Regardless of whether the e-tailer shares or not share demand information, the manufacturer should encroach on the direct channel when the spillover effect is low. When the spillover effect is moderate, the manufacturer should encroach on the direct channel if the marginal cost of DDM effort is relatively low or large, and the manufacturer should not encroach on the direct channel if the marginal cost of DDM effort is moderate. When the spillover effect is large, the manufacturer should not encroach on the direct channel if the marginal cost of DDM effort is low, and otherwise. When the manufacturer encroaches on the direct channel and the online retailer shares information, there exists a win-win situation for both the manufacturer and the online retailer when the spillover effect and the marginal cost of DDM effort are all low. We extend the base case to the case that the manufacturer can adopt the agency channel, and we show that the manufacturer should adopt the agency channel only if the marginal cost of DDM effort is moderate and the commission rate is low.

The remainder of the paper is organized as follows. In the next section, we position our paper in the context of the literature. In section 3, we present the model assumption, demand functions, and game sequence. In section 4, we derive the equilibrium outcomes under different scenarios and make the comparative analysis. In section 5, we extend the model to the case that the manufacturer can select the agency channel. In section 6, we present our conclusion. The proofs of all the propositions are presented in the online appendix S1 File.

## 2. Literature review

Our work is closely related to three streams of literature: supplier encroachment and dual-channel retailing, vertical information sharing in the supply chain, and Data-driven Marketing in Retail Management. In this section, we provide an overview of these research streams and highlight our contributions to the related work.

### 2.1. Supplier encroachment and dual-channel retailing

Dual-channel retailing has attracted much attention over the past decades, and many scholars have studied the manufacturer's channel selection decisions, including whether the manufacturer should introduce an online channel in addition to the traditional reselling channel to sell products directly to consumers. The behavior of the manufacturer competing with the retailer by introducing a new channel is known as supplier encroachment, which is a common way for manufacturers to increase online sales [3,4]. Huang et al. [5] consider two key features of the hybrid selling strategy are captured: the demand expansion and profit margin effects. Yang et al. [6] consider nonlinear pricing and show that supplier encroachment always hurts the retailer's profit, and the impact on the manufacturer itself depends on the manufacturer's bargaining power. However, there are also some studies showing that supplier encroachment can be beneficial to the retailer. Tsay and Agrawal [7] show that introducing the direct channel by the manufacturer can promote greater profit for both the manufacturer and the retailer when the reselling channel is less efficient in stimulating demand through sales effort. Arya et al. [8] show that when the online direct channel is relatively cost-inefficient, the manufacturer will lower the wholesale price to drive consumers to the retail channel, and there will be a win-win situation for both the manufacturer and the retailer when the wholesale price effect is significant. Wang et al. [9] explore channel selection and pricing strategies in the supply chain and find that multichannel sales are the best option for the manufacturer when the difference in operating costs between online and offline channels is low enough. Shen et al. [10] examine the channel choices for the manufacturer to sell products through the online agency channel or the brick-and-mortar store. Zhang and Hezarkhani [11] investigate the manufacturer's channel choice decisions among the direct channel, the reselling channel, and the dual channel. There are also some scholars working on pricing strategies [12,13] and product distribution strategies [14,15] in the background of dual-channel competition. Our paper adds to this literature by analyzing the interplay between information sharing and encroachment and investigating the impact of the spillover effect and the cost of DDM effort on the decisions.

### 2.2. Vertical information sharing in the supply chain

Our study belongs to the vast literature on vertical information sharing in supply chains [16–19]. In this stream of research, many studies consider the situation where the downstream retailer has more demand information than the supplier and analyze the information sharing incentives. Ha and Tong [20] and Ha et al. [21] further extend the model to the setting of channel competition, where retailers can share information with the manufacturer in each channel. Li and Zhang [22] investigate the effect of information confidentiality on supply chain members' incentive to share information in a competitive environment that consists of a manufacturer and multiple retailers. Yoon et al. [23] consider a multi-tier supply chain consisting of a manufacturer, a first-tier manufacturer, and a second-tier manufacturer, where the first-tier manufacturer has access to the second-tier manufacturer's information and may share the information with the manufacturer, and they investigate how information sharing strategy affects the manufacturer's and the first-tier manufacturer's decision making. Guan et al. [24] investigate the value of information sharing when the manufacturer encroaches on the retailer's market by introducing a direct channel. Liu et al. [25] investigate the optimal information sharing strategy of the online retail platform while multiple sellers compete. Ha et al. [26] develop a game model to explore the manufacturer's agency channel encroaching strategy and the online platform's information sharing strategy when the reselling channel already exists. Tang et al. [27] study the manufacturer's encroachment strategy (the direct channel or the agency channel) and the platform's information sharing strategy and show that the online platform is more willing to share information to induce

manufacturers to encroach on the agency channel. Cui et al. [28] consider the manufacturer's channel choice (the reselling channel or the agency channel) and the platform's information sharing strategy in the case where the online platform wholesales products from the manufacturer as the private label. To the best of our knowledge, none of these papers examine the impact of information sharing on firms' channel structure decisions in the e-commerce supply chain considering DDM effort.

### 2.3. Data-driven marketing in retail management

A couple of insightful discussion papers related to data-driven marketing have been published in recent years. Most of the literature analyzes DDM qualitatively, including techniques, opportunities, regulation, data foundations, and applications [29–31]. For example, Rust and Huang [32] discuss how big data will revolutionize service research and transform marketing science research. Aral and Walker [33] conducted experiments on social media (Facebook) to randomly manipulate the messages sent by users of a particular feature. The authors determined the impact of bond strength and embeddedness. Their findings illustrate how social analytics can be used to improve marketing operations. Cali and Balaman [34] propose a novel decision support system for product ranking problems. They consider a set of product criteria and customer reviews posted on the website related to these criteria to recommend the most appropriate choice to potential customers. Several scholars have analyzed the impact of DDM on decision-making and profitability by building mathematical models. Ghoshal et al. [35] quantitatively analyzed a platform's data federation decisions in terms of personalized recommendations based on big data. They found that non-personalized companies are always willing to share their data with personalized companies. Liu et al. [36] investigate a platform's preferences between agency selling and reselling considering the impact of DDM. Xu et al. [37] explore how DDM impacts the quality improvement effort, online inventory level, and offline order quantity. Xing et al. [38] analyze different pricing scenarios in the video service supply chain considering DDM. Our paper makes a novel contribution to this literature by showing how channel structure and information sharing impact an e-tailer's incentive to exert DDM effort.

### 3. Modeling framework

We consider a vertical supply chain with a manufacturer (denoted as M) and an e-tailer (denoted as P) with private demand information. The manufacturer sells products in the end market through the e-tailer, and the manufacturer chooses whether or not to encroach on a direct channel to sell products directly to consumers. We assume that products distributed by the reselling channel and the direct channel are perfect substitutes for consumers. Notations are summarized in Table 1.

Because of the uncertainty of product market demand, we assume the market demand $a$ follows the normal distribution $N(a_0, \sigma_a^2)$ known to both the e-tailer and the manufacturer. $\sigma_a^2$ represents the variance of the market size distribution, and a higher $\sigma_a^2$ indicates a greater degree of demand fluctuation. The e-tailer predicts potential demand as $Y$ based on customer-seller interactions, such as the customers' clickstream data. $Y$ is an unbiased estimator of $a$, and $Y$ is given by $Y = a + \varepsilon$, where the random variable $\varepsilon$ represents the uncertainty level for the market potential, which follows the normal distribution $N(a_0, \sigma_\varepsilon^2)$. Therefore, $Y$ follows the normal distribution $N(a_0, \sigma_a^2 + \sigma_\varepsilon^2)$, so we denote $Y = a_0 + \xi\sqrt{\sigma_a^2 + \sigma_\varepsilon^2}$, where $\xi = \frac{(Y-a_0)}{\sqrt{\sigma_a^2 + \sigma_\varepsilon^2}}$, i.e., the standardized demand signal, $\xi \sim N(0, 1)$. According to the covariance property of the normal distribution, the posterior probability of market demand $a$ with respect to the standardized demand signal $\xi$ also follows the normal distribution, i.e., $a|\xi \sim N(a_0 + \xi\sigma_a\sqrt{1-\sigma^2}, \sigma_a^2\sigma^2)$, where $\sigma^2 = \sigma_\varepsilon^2/(\sigma_a^2 + \sigma_\varepsilon^2)$, and $1-\sigma^2$ can be viewed as the degree of accuracy of the e-tailer's demand forecasting. The e-tailer can know the actual demand for products in the market, whereas the manufacturer only knows the distribution of market demand, and the e-tailer shares the information about market demand exclusively with the manufacturer when an information-sharing contract is reached. We assume that the reselling channel and the direct channel have the same marginal operating cost [8], and without loss of generality,

**Table 1. Summary of notation.**

| Notation | Description |
|---|---|
| NN | No direct channel and no information sharing |
| NS | No direct channel and information sharing |
| EN | Direct channel and no information sharing |
| ES | Direct channel and information sharing |
| a | Potential market size |
| $\xi$ | Standardized demand signal |
| w | Wholesale price |
| $q_R$ | Selling quantity in reselling channel |
| $q_M$ | Selling quantity in direct channel |
| e | DDM effort |
| k | Marginal cost coefficient of DDM effort |
| $\eta$ | Spillover effect coefficient |
| $\Pi_M$ | Profit of the manufacturer |
| $\Pi_P$ | Profit of the e-tailer |
| * | Superscript of equilibrium solutions |

the manufacturer's marginal production cost and the marginal operating cost of the channels are normalized to zero. When the manufacturer encroaches on the direct channel, the manufacturer and the e-tailer compete on quantities, and considering the change in consumer demand due to the e-tailer's DDM effort, the product prices for each channel derived from the inverse demand function are respectively:

$$p_R = a + e - q_R - q_M \tag{1}$$

$$p_M = a + \eta e - q_M - q_R \tag{2}$$

where $q_R$ and $q_M$ represent the e-tailer's selling quantity in the reselling channel and the manufacturer's selling quantity in the direct channel, respectively, and $e$ is the e-tailer's DDM effort, which can be interpreted as short-term marketing campaigns by the e-tailer to boost demand. The increase in the brand image and product awareness resulting from DDM effort is not limited to a single channel, and the e-tailers' DDM effort also can positively affect the demand in the manufacturer's direct channel, and we denote $\eta \in (0, 1)$ as the spillover effect coefficient of the DDM effort, which measures the spillover effect of the e-tailers' DDM effort on the manufacturer's direct channel. If the manufacturer does not encroach on the direct channel, $q_M$ is 0. Fig 1 shows the supply chain structure under the four decision scenarios.

The cost of market effort incurred by the e-tailer is given by $ke^2/2$ where a higher $k$ means a higher cost of market effort. Here, a quadratic cost function implies that it is increasingly more costly to exert DDM effort to achieve a unit increase in demand $e$. The timing of the game is illustrated in Fig 2.

The sequence of events is as follows. In Stage 1, the e-tailer chooses whether or not to share information. In Stage 2, the manufacturer chooses whether or not to encroach on the direct channel. In Stage 3, the manufacturer sets the wholesale price $w$. In Stage 4, the e-tailer determines the selling quantity $q_R$ for the reselling channel. In Stage 5, the e-tailer determines DDM effort $e$, and the manufacturer decides on the selling quantity $q_M$ simultaneously if the manufacturer encroaches on the direct channel. In the last stage (Stage 6), the market price is realized, and then the manufacturer and the e-tailer receive their payoffs.

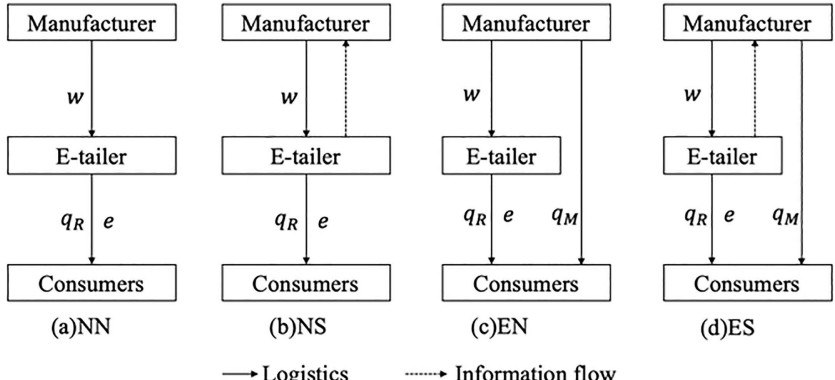

**Fig 1. Supply chain structure under the four decision scenarios.**

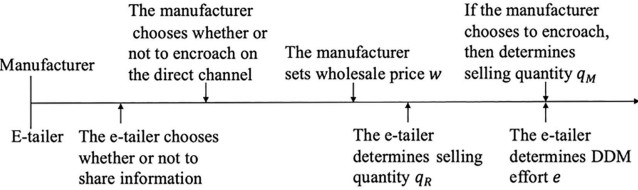

**Fig 2. Timing of the game.**

## 4. Model analysis

We consider four scenarios in which the manufacturer does or does not encroach on the direct channel and the e-tailer does or does not share market demand information, using superscripts to indicate the four scenarios: no direct channel and no information sharing (NN), no direct channel and information sharing (NS), direct channel and no Information sharing (EN), and direct channel and information sharing (ES). To ensure the profit function is concave and the selling quantity of each channel is positive, we assume $k > k_{min} = max\{1/2, (2-3\eta+2\sqrt{1-3(1-\eta)\eta})/3, 1-\eta, (4-5\eta)/3\}$.

### 4.1. Scenario NN: no direct channel and no information sharing

In this setting, the expected profits of the manufacturer and the e-tailer are, respectively

$$\Pi_M^{NN} = wq_R \tag{3}$$

$$\Pi_P^{NN} = ((a_0 + \xi\sigma_a\sqrt{1-\sigma^2}) + e - q_R - w)q_R - \frac{k}{2}e^2 \tag{4}$$

With backward induction, given the wholesale price of the product $w$, the e-tailer solves the following problem

$$\max_{e,q_R}((a_0 + \xi\sigma_a\sqrt{1-\sigma^2}) + e - q_R - w)q_R - \frac{k}{2}e^2 \tag{5}$$

By solving the above problem, we derive the e-tailer's best responses to wholesale price $w$:

$$q_R = \frac{k(a_0 - w + \xi\sqrt{1-\sigma^2}\sigma_a)}{2k-1} \tag{6}$$

$$e = \frac{a_0 - w + \xi\sqrt{1-\sigma^2}\sigma_a}{2k-1} \tag{7}$$

Without information sharing, by anticipating the e-tailer's best-response functions, the manufacturer solves the following problem

$$\max_w \frac{kw(a_0 - w)}{2k-1} \tag{8}$$

By solving the above optimization problem, we obtain the equilibrium wholesale price $w^{NN*}$:

$$w^{NN*} = \frac{a_0}{2} \tag{9}$$

By anticipating $w^{NN*}$ into the e-tailer's best-response functions, we have the following equilibrium decisions:

$$q_R^{NN*} = \frac{k(a_0 + 2\xi\sqrt{1-\sigma^2}\sigma_a)}{2(2k-1)} \tag{10}$$

$$e^{NN*} = \frac{a_0 + 2\xi\sqrt{1-\sigma^2}\sigma_a}{2(2k-1)} \tag{11}$$

By anticipating the above equilibrium results back to the profit functions, we have

$$\Pi_M^{NN*} = \frac{ka_0^2}{4(2k-1)} \tag{12}$$

$$\Pi_P^{NN*} = \frac{k(a_0^2 + 4(1-\sigma^2)\sigma_a^2)}{8(2k-1)} \tag{13}$$

The above equilibrium results demonstrate that the manufacturer's wholesale price and profit are independent of the demand signal when the e-tailer does not share information, and the manufacturer can only make decisions based on the prior beliefs. The e-tailer has an information advantage, the greater the precision of the information, the more efficiently the e-tailer can cope with the uncertainty of the market information, and the greater the selling quantity, the sells effort, and the profit, that is, $\frac{\partial q_R^{NN*}}{\partial(1-\sigma^2)} > 0, \frac{\partial e^{NN*}}{\partial(1-\sigma^2)} > 0, \frac{\partial \Pi_P^{NN*}}{\partial(1-\sigma^2)} > 0$.

### 4.2. Scenario NS: no direct channel and information sharing

In this setting, given the wholesale price of the product $w$, the e-tailer solves the following problem

$$\max_{e,q_R}((a_0 + \xi\sigma_a\sqrt{1-\sigma^2}) + e - q_R - w)q_R - \frac{k}{2}e^2 \tag{14}$$

By solving the above problem, we derive the e-tailer's best responses to wholesale price $w$:

$$q_R = \frac{k(a_0 - w + \xi\sqrt{1-\sigma^2}\sigma_a)}{2k-1} \tag{15}$$

$$e = \frac{a_0 - w + \xi\sqrt{1-\sigma^2}\sigma_a}{2k-1} \tag{16}$$

With information sharing, based on the e-tailer's best-response functions, the manufacturer solves the following problem

$$\max_w \frac{kw(a_0 - w + \xi\sqrt{1-\sigma^2}\sigma_a)}{2k-1} \tag{17}$$

By solving the above optimization problem, we solve the equilibrium wholesale price $w^{NS*}$:

$$w^{NS*} = \frac{1}{2}(a_0 + \xi\sqrt{1-\sigma^2}\sigma_a) \tag{18}$$

By anticipating $w^{NS*}$ into the e-tailer's best-response functions, we have the following equilibrium decisions:

$$q_R^{NS*} = \frac{k(a_0 + \xi\sqrt{1-\sigma^2}\sigma_a)}{2(2k-1)} \tag{19}$$

$$e^{NS*} = \frac{a_0 + \xi\sqrt{1-\sigma^2}\sigma_a}{2(2k-1)} \tag{20}$$

By anticipating the above equilibrium results back to the profit functions, we have the manufacturer's and e-tailer's ex ante profits are given by:

$$\Pi_M^{NS*} = \frac{k(a_0^2 + (1-\sigma^2)\sigma_a^2)}{4(2k-1)} \tag{21}$$

$$\Pi_P^{NS*} = \frac{k(a_0^2 + (1-\sigma^2)\sigma_a^2)}{8(2k-1)} \tag{22}$$

The above equilibrium results demonstrate that the manufacturer's wholesale price and profit, as well as the e-tailer's DDM effort, profit, and selling quantity increase with information precision when the e-tailer share information.

### 4.3. Scenario EN: direct channel and no information sharing

In this setting, the manufacturer and the e-tailer solve the following problems

$$\max_e ((a_0 + \xi\sigma_a\sqrt{1-\sigma^2}) + e - q_R - q_M - w)q_R - \frac{k}{2}e^2 \tag{23}$$

$$\max_{q_M} wq_R + (a_0 + \eta e - q_M - q_R)q_M \tag{24}$$

By solving the above problem, we derive the manufacturer's and e-tailer's best response to selling quantity $q_R$ in the reselling channel:

$$q_M = \frac{ka_0 - (k - \eta)q_R}{2k} \tag{25}$$

$$e = \frac{q_R}{k} \tag{26}$$

Without information sharing, taking the above best response functions into account, the e-tailer solves the following problem

$$\max_{q_R} \frac{q_R(ka_0 - (k + \eta - 1))q_R - 2kw + 2k\xi\sqrt{1 - \sigma^2}\sigma_a)}{2k} \tag{27}$$

By solving the above optimization problem, we derive the e-tailer's best response with respect to wholesale price $w$:

$$q_R = \frac{k(a_0 - 2w + 2\xi\sqrt{1 - \sigma^2}\sigma_a)}{2(k + \eta - 1)} \tag{28}$$

By anticipating the e-tailer's best response function into the manufacturer's profit function, the manufacturer solves the following problem:

$$\max_w \frac{1}{16(k + \eta - 1)^2}((k + 3\eta - 2)^2 a_0^2 + 4(4k - 3k^2 - 6k\eta + \eta^2)w^2 + 4a_0((3k^2 - 4k(1 - \eta) + (2 - 3\eta)\eta)w)) \tag{29}$$

By solving the above optimization problem, we solve the equilibrium wholesale price $w^{EN*}$:

$$w^{EN*} = \frac{(k(3k - 4) + 6k\eta - \eta^2 - 2\eta(k + \eta - 1))a_0}{2(k(3k - 4) + 6k\eta - \eta^2)} \tag{30}$$

By anticipating $w^{NN*}$ back, we have the following equilibrium decisions:

$$q_M^{EN*} = \frac{k(3k + 5\eta - 4)a_0}{2(k(3k - 4) + 6k\eta - \eta^2)} \tag{31}$$

$$q_R^{EN*} = \frac{k\eta a_0}{k(3k - 4) + 6k\eta - \eta^2} + \frac{k\xi\sqrt{1 - \sigma^2}\sigma_a}{k + \eta - 1} \tag{32}$$

$$e^{EN*} = \frac{\eta a_0}{k(3k - 4) + 6k\eta - \eta^2} + \frac{\xi\sqrt{1 - \sigma^2}\sigma_a}{k + \eta - 1} \tag{33}$$

Consequently, we have

$$\Pi_M^{EN*} = \frac{k(3k + 6\eta - 4)a_0^2}{4(k(3k - 4) + 6k\eta - \eta^2)} \tag{34}$$

$$\Pi_P^{EN*} = \frac{k\eta^2(k+\eta-1)a_0^2}{2(k(3k-4)+6k\eta-\eta^2)^2} - \frac{k(1-2\eta)(1-\sigma^2)\sigma_a^2}{2(k+\eta-1)^2} \tag{35}$$

The above equilibrium results demonstrate that the e-tailer's profit decreases with increasing information precision (i.e., $\frac{\partial \Pi_P^{EN*}}{\partial(1-\sigma^2)} < 0$) when the spillover effect is low ($\eta < \frac{1}{2}$)

### 4.4. Scenario ES: direct channel and information sharing

In this setting, the manufacturer and the e-tailer solve the following problems

$$\max_e ((a_0 + \xi\sigma_a\sqrt{1-\sigma^2}) + e - q_R - q_M - w)q_R - \frac{k}{2}e^2 \tag{36}$$

$$\max_{q_M} wq_R + ((a_0 + \xi\sigma_a\sqrt{1-\sigma^2}) + \eta e - q_M - q_R)q_M \tag{37}$$

As before, we derive the manufacturer's and e-tailer's best response to selling quantity $q_R$ in the reselling channel:

$$q_M = \frac{ka_0 - (k-\eta)q_R + k\xi\sqrt{1-\sigma^2}\sigma_a}{2k} \tag{38}$$

$$e = \frac{q_R}{k} \tag{39}$$

With information sharing, by anticipating the above best-response functions, the e-tailer solves the following problem

$$\max_{q_R} \frac{q_R(ka_0 - (k+\eta-1))q_R - 2kw + k\xi\sqrt{1-\sigma^2}\sigma_a)}{2k} \tag{40}$$

By solving the above optimization problem, we derive the e-tailer's best response with respect to wholesale price $w$:

$$q_R = \frac{k(a_0 - 2w + \xi\sqrt{1-\sigma^2}\sigma_a)}{2(k+\eta-1)} \tag{41}$$

By taking the e-tailer's best response function into account, the manufacturer solves the following problem:

$$\max_w \frac{1}{16(k+\eta-1)^2}((k+3\eta-2)^2a_0^2 + 4(k(4-6\eta)+\eta^2-3k^2)w^2 + 4(3k^2 + 4k(1-\eta)$$
$$+ (2-3\eta)\eta)\xi\sqrt{1-\sigma^2}w\sigma_a + (k+3\eta-2)^2\xi^2(1-\sigma^2)\sigma_a^2 + 2a_0((6k^2 + 8k(1-\eta) + 2(2-3\eta)\eta)w$$
$$+ (k+3\eta-2)^2\xi\sqrt{1-\sigma^2}\sigma_a)) \tag{42}$$

By solving the above optimization problem, we solve the equilibrium wholesale price $w^{ES*}$:

$$w^{ES*} = \frac{(k(3k-4)+6k\eta-\eta^2-2\eta(k+\eta-1))(a_0+\xi\sqrt{1-\sigma^2}\sigma_a)}{2(k(3k-4)+6k\eta-\eta^2)} \tag{43}$$

By substituteing $w^{NN*}$ back, we have the following equilibrium decisions:

$$q_M^{ES*} = \frac{k(3k+5\eta-4)(a_0+\xi\sqrt{1-\sigma^2}\sigma_a)}{2(k(3k-4)+6k\eta-\eta^2)}$$

(44)

$$q_R^{ES*} = \frac{k\eta(a_0+\xi\sqrt{1-\sigma^2}\sigma_a)}{k(3k-4)+6k\eta-\eta^2}$$

(45)

$$e^{ES*} = \frac{\eta(a_0+\xi\sqrt{1-\sigma^2}\sigma_a)}{k(3k-4)+6k\eta-\eta^2}$$

(46)

By substituteing the above equilibrium decisions back to the profit functions, we have

$$\Pi_M^{ES*} = \frac{k(3k+6\eta-4)(a_0^2+(1-\sigma^2)\sigma_a^2)}{4(k(3k-4)+6k\eta-\eta^2)}$$

(47)

$$\Pi_P^{ES*} = \frac{k\eta^2(k+\eta-1)(a_0^2+(1-\sigma^2)\sigma_a^2)}{2(k(3k-4)+6k\eta-\eta^2)^2}$$

(48)

The above equilibrium results demonstrate that the equilibrium decisions increase with the increase of information precision.

## 4.5. Comparative analysis

We compare the equilibrium DDM effort, equilibrium wholesale price, equilibrium selling quantity, and the equilibrium profits of the manufacturer and the e-tailer in scenario NN and scenario NS leading to Proposition 1.

**Proposition 1.** *(a)* $e^{NS*} < e^{NN*}$. *(b)* $w^{NS*} > w^{NN*}$. *(c)* $q_R^{NS*} < q_R^{NN*}$. *(d)* $\Pi_M^{NS*} > \Pi_M^{NN*}$, *and* $\Pi_P^{NS*} < \Pi_P^{NN*}$.

Proposition 1 demonstrates that when there is no encroachment, if the e-tailer shares information, then the manufacturer adjusts the wholesale price based on demand information, which enhances the double marginal effect of the wholesale price, making the platform adjust the retail price consequently, leading to higher retail price and lower selling quantity. Information sharing by the e-tailer is beneficial to the manufacturer but lowers the profitability of the e-tailer, and the above findings are inconsistent with the existing literature [26]. Part (a) shows that when the e-tailer shares demand information, it becomes less profitable for the e-tailer and the e-tailer has less incentive to conduct DDM effort. We compare the equilibrium DDM effort, equilibrium wholesale price, equilibrium selling quantity, and the equilibrium profits of the manufacturer and the e-tailer in scenario EN and scenario ES leading to Proposition 2.

**Proposition 2.** *(a)* $e^{ES*} < e^{EN*}$. *(b) There exists* $k_1$ *such that* $w^{ES*} > w^{EN*}$ *if (i)* $0 < \eta < \frac{5}{6}$ *or (ii)* $\frac{5}{6} < \eta < 1$ *and* $k > k_1$, *and* $w^{ES*} < w^{EN*}$ *if* $\frac{5}{6} < \eta < 1$ *and* $k > k_1$, *where* $k_1$ *is given in the Appendix B in* <u>S1 File</u>. *(c)* $q_R^{ES*} < q_R^{EN*}$. $q_M^{ES*} > q_M^{EN*}$. *(d)* $\Pi_M^{ES*} > \Pi_M^{EN*}$. *When* $0 < \eta < \frac{1}{2}$, $\Pi_P^{ES*} > \Pi_P^{EN*}$ , *and* $\Pi_P^{ES*} < \Pi_P^{EN*}$ *otherwise.*

Proposition 2 shows that when the manufacturer encroaches on the direct channel if the e-tailer shares information, it leads to a decrease in selling quantity in the reselling channel, an increase in selling quantity in the direct channel, and a decrease in DDM effort. Part (b) shows that the manufacturer may increase the wholesale price of the product when the spillover effect is low or when the spillover effect is large and the marginal cost of DDM effort is large. The manufacturer has the incentive to reduce the wholesale price to incentivize the e-tailer to increase the DDM effort when the spillover effect is large and the marginal cost of DDM effort is low. This can be explained by the manufacturer's incentive

to motivate the e-tailer to exert more DDM effort. Fig 3 illustrates the results. Part (d) shows that information sharing is beneficial to the manufacturer and the e-tailer may share information when the spillover effect is low.

Now, we compare the equilibrium DDM effort, equilibrium wholesale price, and equilibrium selling quantity in scenario EN and scenario NN leading to Proposition 3.

**Proposition 3.** *(a) There exists $k_2$ and $a_1$ such that $e^{EN*} > e^{NN*}$ if (i) $0 < \eta < \frac{1}{2}$ and $k_{min} < k < k_2$ or (ii) $0 < \eta < \frac{1}{2}$, $a_0 < a_1$ and $k > k_2$ or (iii) $\frac{1}{2} < \eta < 1$, $a_0 < a_1$ and $k > \eta$, and $e^{EN*} < e^{NN*}$ if (i) $0 < \eta < \frac{1}{2}$, $a_0 > a_1$ and $k > k_2$ or (ii) $\frac{1}{2} < \eta < 1$ and $k_{min} < k < \eta$ or (iii) $\frac{1}{2} < \eta < 1$, $a_0 > a_1$ and $k > \eta$, where $a_1$ and $k_2$ are given in the Appendix B in* S1 File. *(b) $w^{EN*} < w^{NN*}$.*

Proposition 3 implies when the e-tailer does not share information, if the manufacturer encroaches on the direct channel, the manufacturer reduces the wholesale price. The e-tailer has an information advantage over the manufacturer and knows the actual demand for products in the market. If the marginal cost of DDM effort is low, the e-tailer has more incentive to exert DDM effort when the spillover effect is low and decreases DDM effort when the spillover effect is large. The e-tailer should increase DDM effort when the mean market size is lower than a threshold, and decrease DDM effort when the mean market size is larger than a threshold. We compare the equilibrium DDM effort and equilibrium wholesale price in scenario NS and scenario ES leading to Proposition 4.

**Proposition 4.** *(a) $e^{ES*} > e^{NS*}$ if (i) $0 < \eta < \frac{1}{2}$ and $k_{min} < k < k_2$ or (ii) $\frac{1}{2} < \eta < 1$, and $e^{ES*} < e^{NS*}$ if $0 < \eta < \frac{1}{2}$ and $k > k_2$. (b) $w^{ES*} < w^{NS*}$.*

Proposition 4 demonstrates when the e-tailer shares information, the manufacturer will reduce the wholesale price if the manufacturer encroaches on the direct channel, the finding is consistent with the case when the e-tailer does not share information. When the spillover effect is low, the e-tailer should increase the DDM effort if the marginal cost of DDM effort is lower than a threshold, decrease the DDM effort if the marginal cost of DDM effort is larger than a threshold, and always decrease the DDM effort when the spillover effect is large. As illustrated in Fig 4, the threshold decreases with the increase in spillover effect

We compare the equilibrium DDM effort and equilibrium wholesale price in scenario NS and scenario ES leading to Proposition 5.

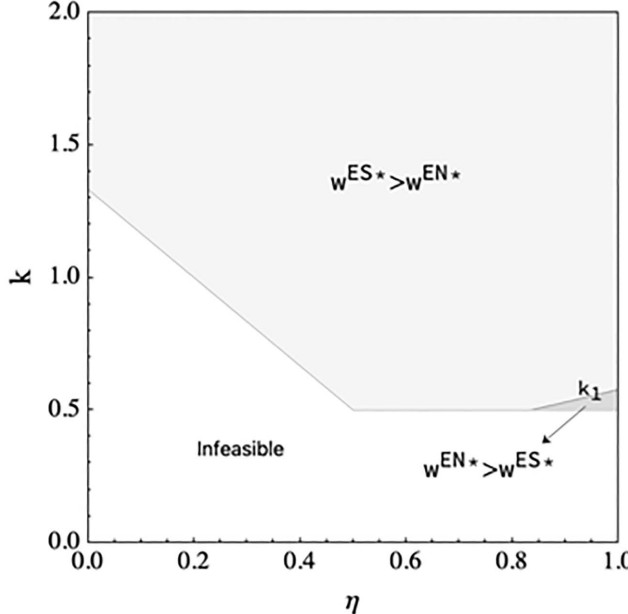

**Fig 3. The wholesale price comparison in scenario EN and scenario ES.**

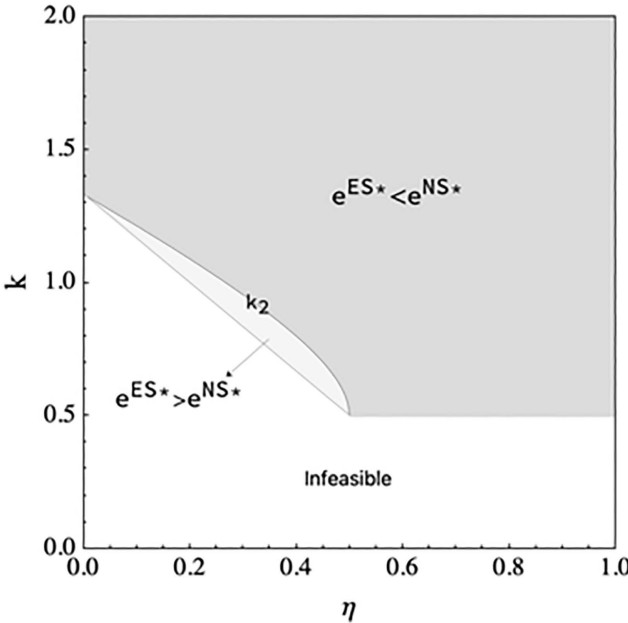

**Fig 4.  The DDM effort comparison in scenario NS and scenario ES.**

**Proposition 5.** *There exists $k_3$ and $k_4$ such that $\Pi_M^{EN*} > \Pi_M^{NN*}$, $\Pi_M^{ES*} > \Pi_M^{NS*}$ if (i) $0 < \eta < \frac{1}{12}(3 + \sqrt{3})$ or (ii) $\frac{1}{12}(3 + \sqrt{3}) < \eta < \frac{1}{2}$ and $k_{min} < k < k_3$ or $k > k_4$ or (iii) $\frac{1}{2} < \eta < 1$ and $k > k_4$, and $\Pi_M^{EN*} < \Pi_M^{NN*}$, $\Pi_M^{ES*} < \Pi_M^{NS*}$ if (i) $\frac{1}{12}(3 + \sqrt{3}) < \eta < \frac{1}{2}$ and $k_3 < k < k_4$ or (ii) $\frac{1}{2} < \eta < 1$ and $k_{min} < k < k_4$, where $k_3$ and $k_4$ are given in the Appendix B in* S1 File*.*

When the manufacturer encroaches on the direct channel, the manufacturer's profit consists of two parts, including the revenue from the reselling channel and the direct channel. when the spillover effect and marginal cost of DDM effort meet certain conditions, the manufacturer's encroachment on the direct channel is not more profitable. From Fig 5, Region A denotes that the manufacturer gains more profit in the case of encroaching on the direct channel than in the case of not encroaching on the direct channel, and Region B denotes that the manufacturer gains less profit in the case of encroaching on the direct channel than in the case of without encroachment. The manufacturer should adjust the selling channel strategy to achieve a higher profit. Regardless of whether the e-tailer shares or not shares demand information the manufacturer should encroach on the direct channel when the spillover effect is low. When the spillover effect is moderate, the manufacturer should encroach on the direct channel if the marginal cost of DDM effort is low or large, and the manufacturer should not encroach on the direct channel if the marginal cost of DDM effort is moderate. When the spillover effect is large, the manufacturer should not encroach on the direct channel if the marginal cost of DDM effort is low, and otherwise

We compare the equilibrium DDM effort and equilibrium wholesale price in scenario NS and scenario ES leading to Proposition 6.

**Proposition 6.** *(a) When the online retail does not share information, there exists $k_5$ and $a_2$ such that $\Pi_P^{EN*} < \Pi_P^{NN*}$ if $0 < \eta < \frac{1}{2}$, $k_{min} < k < k_5$, and $a_0 < a_2$, and $\Pi_P^{EN*} > \Pi_P^{NN*}$ if (i) $0 < \eta < \frac{1}{2}$, $k_{min} < k < k_5$, and $a_0 > a_2$ or (ii) $0 < \eta < \frac{1}{2}$ and $k > k_5$ or (iii) $\frac{1}{2} < \eta < 1$. (b) When the online retail share information, there exists $k_6$ such that $\Pi_P^{ES*} > \Pi_P^{NS*}$ if $0 < \eta < \frac{1}{2}$ and $k_{min} < k < k_6$ and $\Pi_P^{ES*} < \Pi_P^{NS*}$ if (i) $0 < \eta < \frac{1}{2}$ and $k > k_6$ or (ii) $0 < \eta < \frac{1}{2}$, where $k_5$ and $k_6$ are given in the Appendix B in* S1 File*.*

Proposition 6 shows when the e-tailer does not share information and the spillover effect is low, the e-tailer prefers the manufacturer not to encroach on the direct channel if the marginal cost of DDM effort is low and the mean market

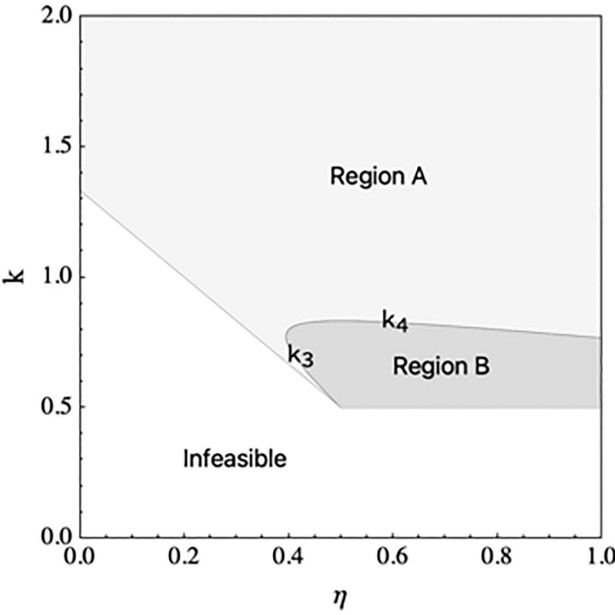

**Fig 5. The manufacturer's profit comparison with or without direct channel.**

size is low, and the e-tailer prefers the manufacturer to encroach on the direct channel if the marginal cost of DDM effort is large or if the marginal cost of DDM effort is low and the mean market size is large. When the spillover effect is large, the e-tailer prefers the manufacturer to encroach on the direct channel. When the e-tailer shares information, when the spillover effect is low, the e-tailer prefers the manufacturer to encroach on the direct channel if the marginal cost of DDM effort is low, and prefers the manufacturer not to encroach on the direct channel if the marginal cost of DDM effort is large. When the spillover effect is large, the e-tailer prefers the manufacturer not to encroach on the direct channel.

**Remark 1.** *When the manufacturer encroaches on the direct channel and the e-tailer shares information, there exists a win-win situation for both the manufacturer and the e-tailer when both the spillover effect and the marginal cost of DDM effort are low.*

## 5. Manufacturer encroach on the agency channel

Many e-tailers started as resellers, who buy products from manufacturers via the reselling channel and then resell to consumers. Over time, some platforms also offer an agency channel that allows manufacturers to sell directly to consumers through their online platforms. For example, Bose sells the products to consumers only via Amazon's reselling and agency channels, whereas Apple distributes the products only through the reselling channel. When the manufacturer encroaches on the agency channel, the product price in the agency channel derived from the inverse demand function is

$$p_M = a + e - q_M - q_R \tag{49}$$

In the agency channel, the manufacturer determines the selling quantity and pays the e-tailer a unit commission rate $\lambda \in (0,1)$ that is proportional to the retail price, and we assume that the commission rate is exogenous. This makes sense because the e-tailer charges the same commission rate for the whole product category and commits to the commission rate before negotiating channel contracts with the manufacturer. We consider two scenarios: the manufacturer encroaches on the agency channel and the e-tailer does not share information (AN), the manufacturer encroaches on the agency channel and the e-tailer shares information (AS). Fig 6 shows the supply chain structure under the two decision scenarios.

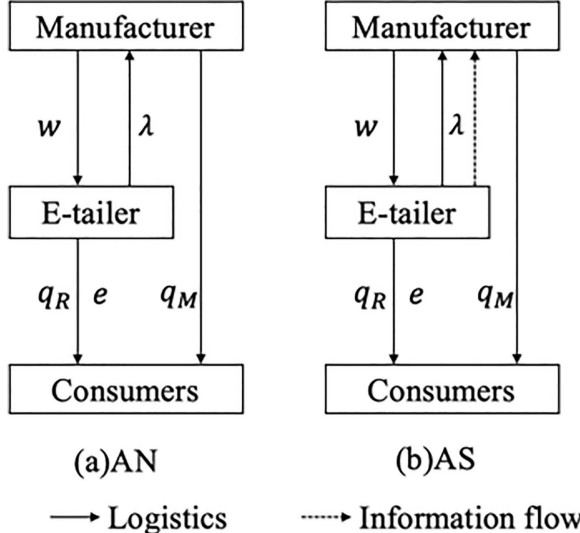

**Fig 6. Supply chain structure under scenario AN and scenario AS.**

When the manufacturer encroaches on the agency channel and the e-tailer does not share information, the expected profits of the manufacturer and the e-tailer are, respectively

$$\max_{e,q_R}\left(\left(a_0 + \xi\sigma_a\sqrt{1-\sigma^2}\right) + e - q_R - q_M - w_R\right)q_R + (1-\lambda)(a_0 + e - q_M - q_R)q_M - \frac{k}{2}e^2 \tag{50}$$

$$\max_{w,q_M} w_R q_R + \lambda(a_0 + e - q_M - q_R)q_M \tag{51}$$

When the manufacturer encroaches on the agency channel and the e-tailer shares information, the expected profits of the manufacturer and the e-tailer are, respectively

$$\max_{e,q_R}\left(\left(a_0 + \xi\sigma_a\sqrt{1-\sigma^2}\right) + e - q_R - q_M - w_R\right)q_R + (1-\lambda)(a_0 + \xi\sigma_a\sqrt{1-\sigma^2} + e - q_M - q_R)q_M - \frac{k}{2}e^2 \tag{52}$$

$$\max_{w,q_M} w_R q_R + \lambda(a_0 + \xi\sigma_a\sqrt{1-\sigma^2} + e - q_M - q_R)q_M \tag{53}$$

Equilibrium results for scenario AN and scenario AS are shown in the Appendix in S1 File. To ensure that the equilibrium solution is non-negative, we obtain $k > \overline{k} = \max\left\{1-\lambda, \frac{1}{3-\lambda}\right\}$. We compare the equilibrium DDM effort and equilibrium wholesale price in scenario AN and scenario AS leading to Proposition 7.

**Proposition 7.** *(a) There exists $k_7$ such that $e^{AS*} < e^{AN*}$ if (i) $0 < \lambda < \frac{1}{2}(5-\sqrt{17})$ or (ii) $\frac{1}{2}(5-\sqrt{17}) < \lambda < 1$ and $k > k_7$ and $e^{AS*} > e^{AN*}$ if $\frac{1}{2}(5-\sqrt{17}) < \lambda < 1$ and $\overline{k} < k < k_7$, where $k_7$ is given in the Appendix B S1 File. (b) $w^{AS*} > w^{AN*}$.*

Proposition 7 demonstrates when the manufacturer encroaches on the agency channel, the manufacturer will increase the wholesale price when the e-tailer shares the demand information. Fig 7 illustrates the results. When the commission rate is low or when the commission rate is high and the marginal cost of DDM effort is large, the e-tailer has more incentive to exert DDM effort if the e-tailer shares demand information, and when the commission rate is high and the marginal cost of DDM effort is low, the e-tailer should decrease DDM effort if the e-tailer shares information.

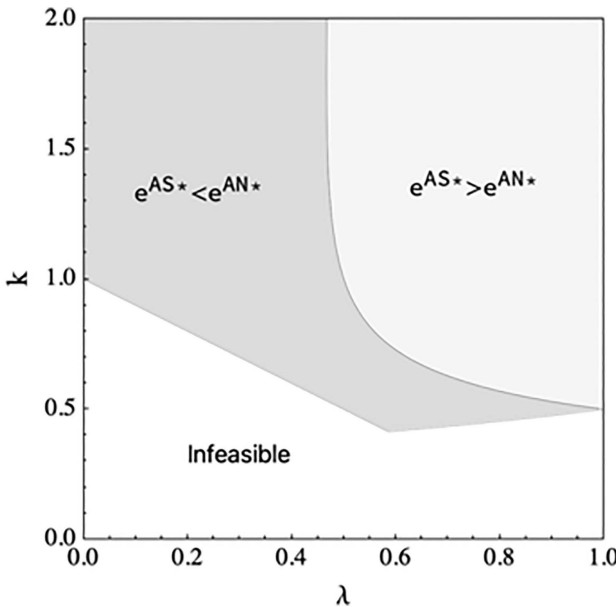

**Fig 7. The DDM effort comparison in scenario AN and scenario AS.**

We analyze the impact of the spillover effect, commission rate, and marginal cost of DDM effort on the DDM effort of the e-tailer in different scenarios. We set $a_0 = 2, \sigma_a = 0.5$, and $\xi = 1$. We allow $k$ to vary form $\max\{k_{min}, \overline{k}\}$ to 2. Fig 8 illustrates the results.

Fig 8 illustrates that when the commission rate is low and the spillover effect is low, the e-tailer has more incentive to exert DDM effort when the manufacturer encroaches on the direct channel if the marginal cost of DDM effort is less than a threshold, and the e-tailer has more incentive to exert DDM effort when the manufacturer does not encroach on another channel if the marginal cost of DDM effort is less than a threshold. When the commission rate is low and the spillover effect is large, the e-tailer has more incentive to exert DDM effort when the manufacturer does not encroach on another channel. When the commission rate is high and the spillover effect is low, the e-tailer has more incentive to exert DDM effort when the manufacturer encroaches on the direct channel if the marginal cost of DDM effort is less than a threshold, and the e-tailer has more incentive to exert DDM effort when the manufacturer encroaches on the agency channel if the marginal cost of DDM effort is less than a threshold. When the commission rate is high and the spillover effect is large, the e-tailer has more incentive to exert DDM effort when the manufacturer encroaches on the agency channel.

We analyze the impact of the spillover effect, commission rate, and marginal cost of DDM effort on the manufacturer's profit in different scenarios. Fig 9 illustrates the results.

Fig 9 shows that when the marginal cost of DDM effort is low, the manufacturer should choose to encroach on the direct channel if the spillover effect is low, and should choose not to encroach on another channel if the spillover effect is large. When the marginal cost of DDM effort is large, the manufacturer should choose to encroach on the direct channel. Intuitively, a lower commission rate benefits the manufacturer because the share of the agency channel's revenue is increased. The results show the manufacturer should choose to encroach on the agency channel only if the marginal cost of DDM effort is moderate and the commission rate is low.

We analyze the spillover effect, the commission rate, and the marginal cost of DDM effort on total supply chain profit under different scenarios. Fig 10 illustrates the results

Fig 10 illustrates that when the marginal cost of DDM effort is large, total supply chain profit is increased when the manufacturer chooses to encroach on the agency channel and the e-tailer shares demand information. When the marginal

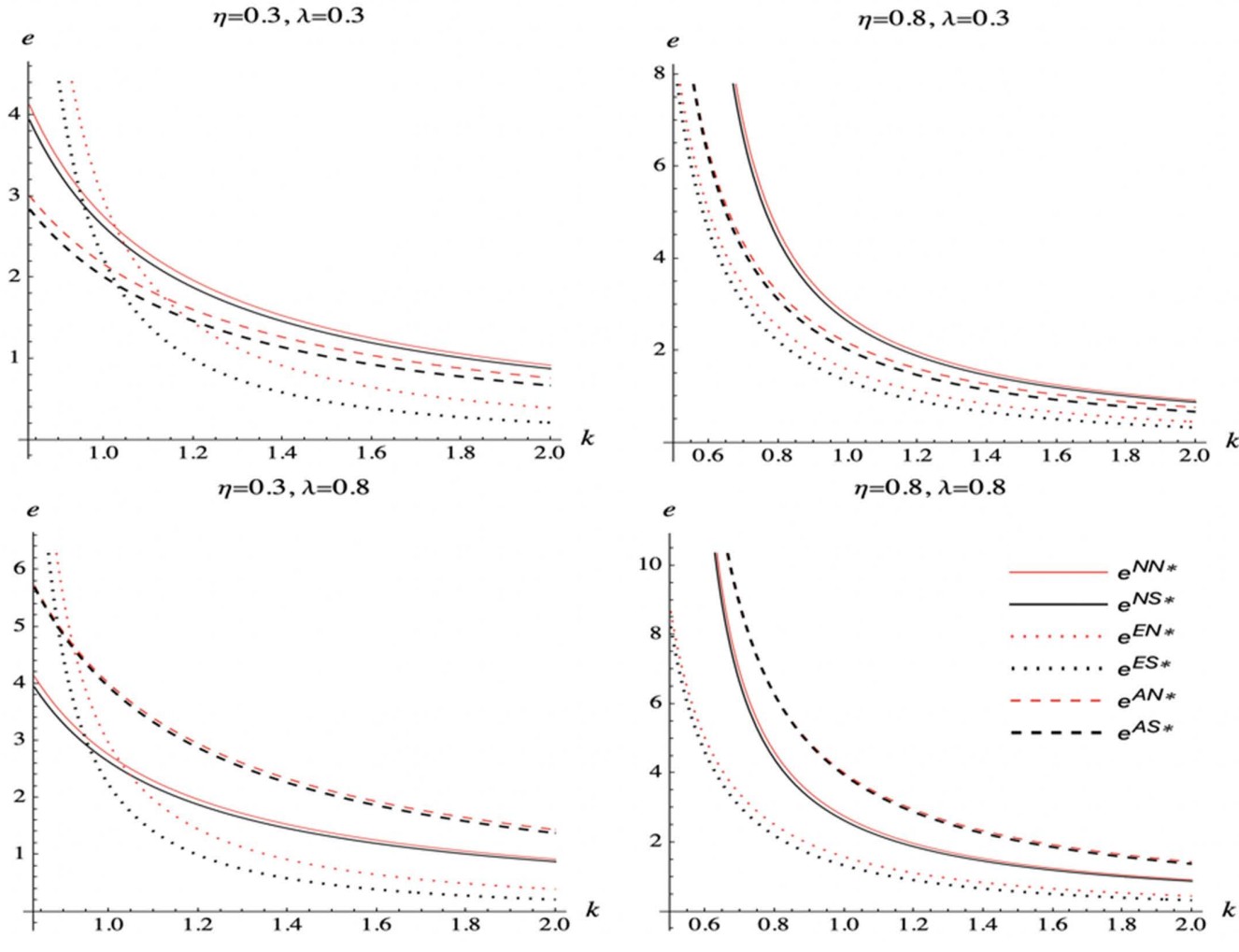

**Fig 8. The e-tailer's DDM effort comparison.**

cost of DDM effort is low, total supply chain profit is increased when the manufacturer does not choose to encroach on another channel and the e-tailer shares demand information if the spillover effect and commission rate are all high.

## 6. Conclusions and practical implications

In this study, we investigate the interplay of information sharing and channel selection decisions in the e-commerce supply chain considering DDM effort. We analyze four scenarios considering the decisions of the manufacturer and the e-tailer: no direct channel and no information sharing (NN), no direct channel and information sharing (NS), direct channel and no Information sharing (EN), and direct channel and information sharing (ES). We compare optimal decisions with manufacturer encroachment channel choices under different supply chain structures. Our analysis reveals that the manufacturer has the incentive to reduce the wholesale price to incentivize the e-tailer to increase the DDM effort when the spillover effect is high and the marginal cost of the DDM effort is low. When the manufacturer encroaches on the direct channel and the online retailer shares information, it leads to a win-win outcome for the manufacturer and the online retailer when both the spillover effect and the marginal cost of DDM effort are low.

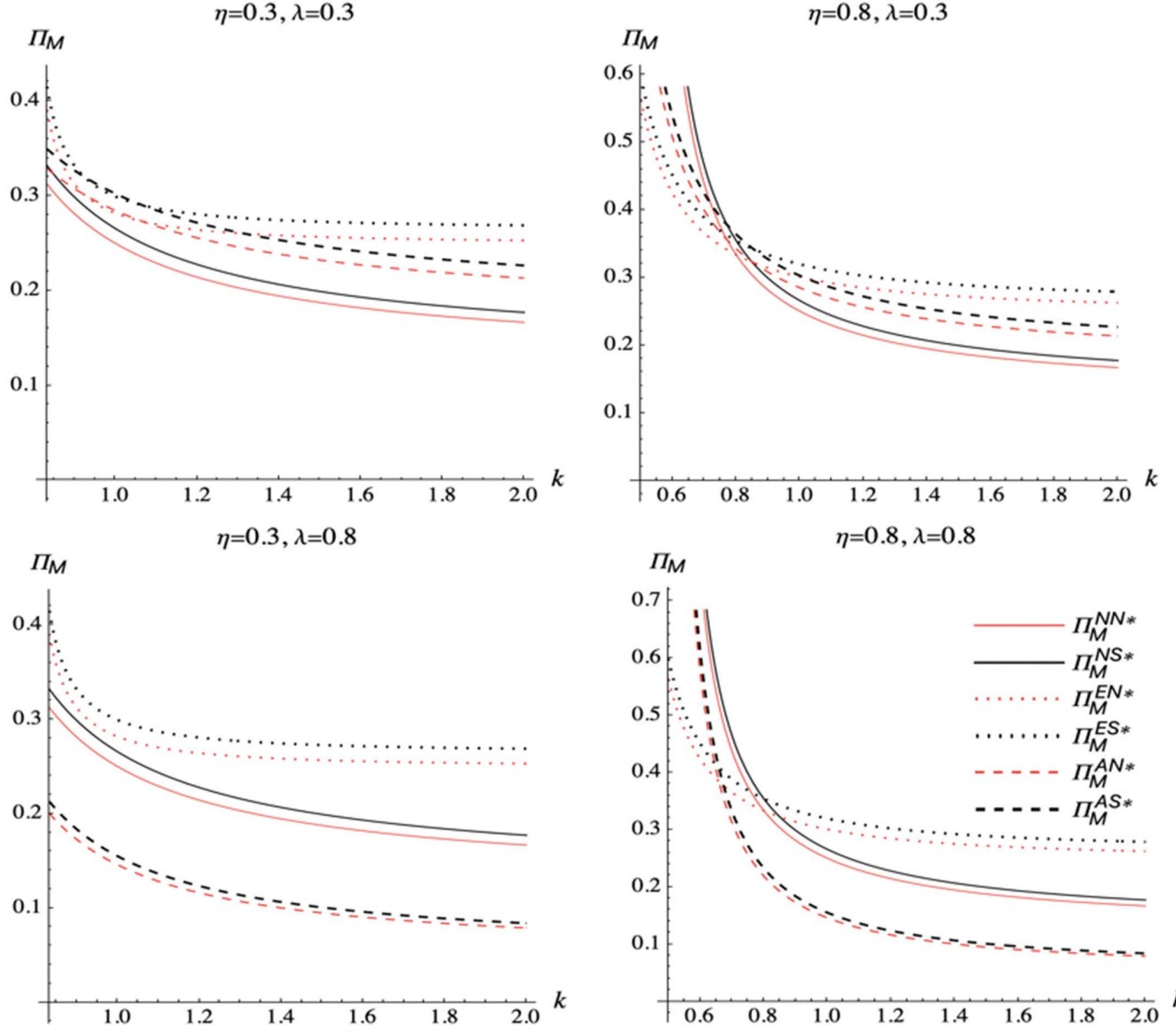

**Fig 9. The manufacturer's profit comparison.**

The findings of this study have important practical implications. When the manufacturer encroaches on the direct channel, the e-tailer may share information if the spillover effect is low. Regardless of whether the e-tailer shares or not shares demand information the manufacturer should encroach on the direct channel when the spillover effect is low. When the spillover effect is moderate, the manufacturer should encroach on the direct channel if the marginal cost of DDM effort is relatively low or high, and the manufacturer should not encroach if the marginal cost of DDM effort is moderate. When the spillover effect is high, the manufacturer should not encroach on the direct channel if the marginal cost of DDM effort is low, and otherwise. We extend the model to the case that the manufacturer can select the agency channel, and we find that the manufacturer should select the agency channel only if the marginal cost of DDM effort is moderate and the commission rate is low. There are several limitations to this research. One limitation of our model is that we assume the

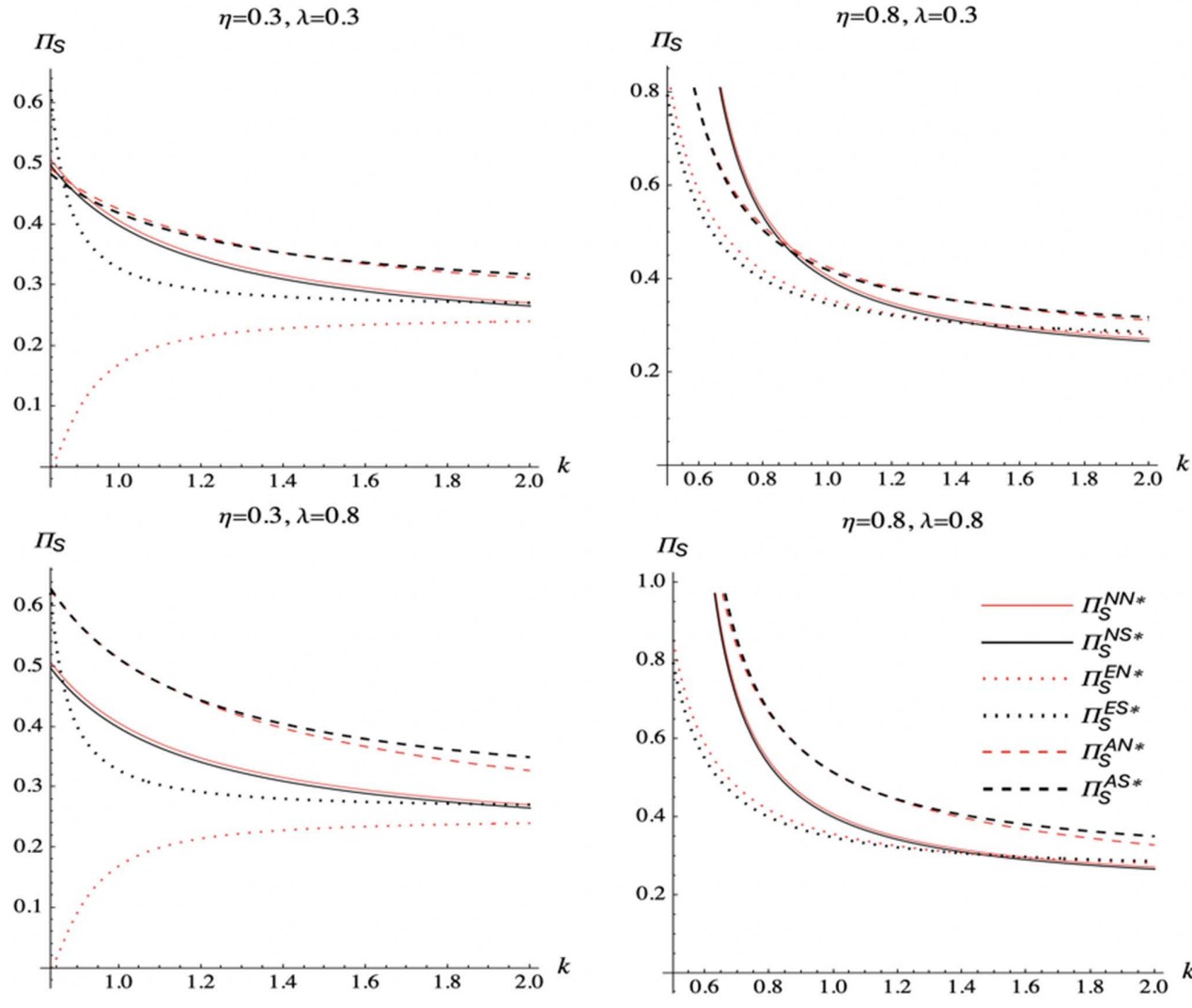

**Fig 10. The total supply chain profit comparison.**

channels to have the same operating cost so that we can focus on some key drivers (e.g., spillover effect and DDM effort). It would be interesting to study how the main findings would change if the operating costs in the channel were different. Another limitation is that we assume that information accuracy is exogenous. It would be worthwhile to incorporate the information acquisition decision into our model.

## Supporting information

**S1 File. Appendix derivation process.**
(PDF)

## Author contributions

**Investigation:** Feifei Han.

**Methodology:** Feifei Han.

**Writing – original draft:** Feifei Han.

**Writing – review & editing:** Mei Wang, Zhengze Wu.

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
