## [Decision Letter · Decision Letter 0]

18 May 2025

Dear Dr. Han,

Thank you for submitting your manuscript to PLOS ONE. After careful consideration, we feel that it has merit but does not fully meet PLOS ONE’s publication criteria as it currently stands. Therefore, we invite you to submit a revised version of the manuscript that addresses the points raised during the review process.

We look forward to receiving your revised manuscript.

Kind regards,

Lisong Zhang

Academic Editor

PLOS ONE

Journal Requirements:

“Shanghai University of Finance and Economics Graduate Innovation Fund (Grant No. CXJJ-2021-379)”

“This research was funded by Shanghai University of Finance and Economics Graduate Innovation Fund (Grant No. CXJJ-2021-379).”

“Shanghai University of Finance and Economics Graduate Innovation Fund (Grant No. CXJJ-2021-379)”

5. We note that your Data Availability Statement is currently as follows: All relevant data are within the manuscript and its Supporting Information files

7. Please ensure that you refer to Figure 6 in your text as, if accepted, production will need this reference to link the reader to the figure.

**Additional Editor Comments:**

Reviewer #1:

The authors develop a game-theoretic model to investigate the encroachment and information sharing decisions considering data-driven marketing develop a game-theoretic model to investigate the encroachment and information sharing decisions considering data-driven marketing. They mathematically derive the results in the paper, and present the derivations in the appendix.

Reviewer 2:

This is a well-written and well-structured manuscript that presents a novel and significant contribution to the field. The methodology is sound, and the results are clearly presented. I recommend it for publication with minor revisions.I suggest the authors address the following points to further strengthen the manuscript:

1.Increase the proportion of recent references — The literature review would benefit from the inclusion of more up-to-date references, particularly from the past two years. This will help situate the current work more clearly within the latest developments in the field.

2.Clarify and highlight the marginal contribution — While the core idea is interesting, the marginal contribution over existing work could be more explicitly emphasized. The authors are encouraged to clearly articulate what distinguishes their approach from closely related studies, both in the introduction and conclusion sections.

Once these points are addressed, I would be happy to recommend the manuscript for publication.

Reviewer 3:

The manuscript is well-organized and clearly written. It provides a thorough and insightful analysis of channel choice and information-sharing decisions in the e-commerce supply chain. The methodology is appropriate, and the results are sound and well-supported. The work represents a valuable contribution to the field.

Recommendation: Accept

Reviewers' comments:

Reviewer's Responses to Questions

**Comments to the Author**

1. Is the manuscript technically sound, and do the data support the conclusions?

Reviewer #1: Yes

Reviewer #2: Yes

Reviewer #3: Yes

2. Has the statistical analysis been performed appropriately and rigorously?

Reviewer #1: Yes

Reviewer #2: Yes

Reviewer #3: Yes

3. Have the authors made all data underlying the findings in their manuscript fully available?

Reviewer #1: Yes

Reviewer #2: Yes

Reviewer #3: Yes

4. Is the manuscript presented in an intelligible fashion and written in standard English?

Reviewer #1: Yes

Reviewer #2: Yes

Reviewer #3: Yes

Reviewer #1: The authors develop a game-theoretic model to investigate the encroachment and information sharing decisions considering data-driven marketing develop a game-theoretic model to investigate the encroachment and information sharing decisions considering data-driven marketing. They mathematically derive the results in the paper, and present the derivations in the appendix.

Reviewer #2: This is a well-written and well-structured manuscript that presents a novel and significant contribution to the field. The methodology is sound, and the results are clearly presented. I recommend it for publication with minor revisions.I suggest the authors address the following points to further strengthen the manuscript:

1.Increase the proportion of recent references — The literature review would benefit from the inclusion of more up-to-date references, particularly from the past two years. This will help situate the current work more clearly within the latest developments in the field.

2.Clarify and highlight the marginal contribution — While the core idea is interesting, the marginal contribution over existing work could be more explicitly emphasized. The authors are encouraged to clearly articulate what distinguishes their approach from closely related studies, both in the introduction and conclusion sections.

Once these points are addressed, I would be happy to recommend the manuscript for publication.

Reviewer #3: The manuscript is well-organized and clearly written. It provides a thorough and insightful analysis of channel choice and information-sharing decisions in the e-commerce supply chain. The methodology is appropriate, and the results are sound and well-supported. The work represents a valuable contribution to the field.

Recommendation: Accept

**Do you want your identity to be public for this peer review?** For information about this choice, including consent withdrawal, please see our Privacy Policy

Reviewer #1: No

Reviewer #2: No

Reviewer #3: No

---

## [Author Response · Author response to Decision Letter 1]

23 Jun 2025

Dear Editor and Reviewers:

Thank you very much for the valuable comments and suggestions regarding our manuscript entitled “Information Sharing and Channel Structure in E-Commerce Supply Chain Considering Data-driven Marketing” (Manuscript ID: PONE-D-25-21171R1). We greatly appreciate the editor and the reviewers’ insightful feedback, which has been extremely helpful in guiding our revisions and improving the overall quality of the manuscript. We have carefully considered all comments and have made the corresponding revisions accordingly. Furthermore, we would like to show the details as follows.

Journal Requirements:

The author’s answer: The manuscript has been revised according to the PLOS ONE style templates as requested.

The author’s answer: There is no author-generated code included in the manuscript.

“Shanghai University of Finance and Economics Graduate Innovation Fund (Grant No. CXJJ-2021-379)”

The author’s answer: The funders had no role in study design, data collection and analysis, decision to publish, or preparation of the manuscript.

“This research was funded by Shanghai University of Finance and Economics Graduate Innovation Fund (Grant No. CXJJ-2021-379).”

“Shanghai University of Finance and Economics Graduate Innovation Fund (Grant No. CXJJ-2021-379)”

The author’s answer: We have removed all funding-related text from the manuscript.

The Funding Statement now reads as follows:

“Zhejiang Provincial Regular Subjects of Philosophy and Social Science Planning (25NDJC104YB)”

5. We note that your Data Availability Statement is currently as follows: All relevant data are within the manuscript and its Supporting Information files

The author’s answer: No new data were generated or analyzed in this study; therefore, data sharing is not applicable to this article.

The author’s answer: No new data were created or analyzed in this study. Data sharing is not applicable to this article.

7. Please ensure that you refer to Figure 6 in your text as, if accepted, production will need this reference to link the reader to the figure.

The author’s answer: We have added a description of Figure 6 to the manuscript.

The author’s answer: We have reviewed the reference list to ensure that it is complete and accurate..

Additional Editor Comments:

Reviewer #1:

The authors develop a game-theoretic model to investigate the encroachment and information sharing decisions considering data-driven marketing develop a game-theoretic model to investigate the encroachment and information sharing decisions considering data-driven marketing. They mathematically derive the results in the paper, and present the derivations in the appendix.

Reviewer 2:

This is a well-written and well-structured manuscript that presents a novel and significant contribution to the field. The methodology is sound, and the results are clearly presented. I recommend it for publication with minor revisions.I suggest the authors address the following points to further strengthen the manuscript:

1.Increase the proportion of recent references — The literature review would benefit from the inclusion of more up-to-date references, particularly from the past two years. This will help situate the current work more clearly within the latest developments in the field.

2.Clarify and highlight the marginal contribution — While the core idea is interesting, the marginal contribution over existing work could be more explicitly emphasized. The authors are encouraged to clearly articulate what distinguishes their approach from closely related studies, both in the introduction and conclusion sections.

Once these points are addressed, I would be happy to recommend the manuscript for publication.

Reviewer 3:

The manuscript is well-organized and clearly written. It provides a thorough and insightful analysis of channel choice and information-sharing decisions in the e-commerce supply chain. The methodology is appropriate, and the results are sound and well-supported. The work represents a valuable contribution to the field.

The author’s answer: We have updated the data in the background section to reflect the most recent available year and increased the proportion of recent references. Additionally, we have added a description of the marginal contribution in the Introduction section.

We would like to thank the referee again for taking the time to review our manuscript.

Yours sincerely,

Feifei Han

---

## [Editor Report · Decision Letter 1]

26 Jun 2025

Information Sharing and Channel Structure in E-Commerce Supply Chain Considering Data-driven Marketing

PONE-D-25-21171R1

Dear Dr. Han,

We’re pleased to inform you that your manuscript has been judged scientifically suitable for publication and will be formally accepted for publication once it meets all outstanding technical requirements.

Kind regards,

Lisong Zhang

Academic Editor

PLOS ONE
---

## [Editor Report · Acceptance letter]

PONE-D-25-21171R1

PLOS ONE

Dear Dr. Han,

I'm pleased to inform you that your manuscript has been deemed suitable for publication in PLOS ONE. Congratulations! Your manuscript is now being handed over to our production team.

Kind regards,

on behalf of

Associate Professor Lisong Zhang

Academic Editor

PLOS ONE